# *Baccharis dracunculifolia* DC Consumption Improves Nociceptive and Depressive-like Behavior in Rats with Experimental Osteoarthritis

**DOI:** 10.3390/foods13040535

**Published:** 2024-02-09

**Authors:** Inês Martins Laranjeira, Elisabete Apolinário, Diana Amorim, Ademar Alves da Silva Filho, Alberto Carlos Pires Dias, Filipa Pinto-Ribeiro

**Affiliations:** 1Life and Health Sciences Research Institute (ICVS), School of Medicine, University of Minho, Campus of Gualtar, 4710-057 Braga, Portugal; inesmartinslaranjeira@gmail.com (I.M.L.); elis91@live.com.pt (E.A.); dianaamorim@med.uminho.pt (D.A.); 2ICVS/3B’s—PT Government Associate Laboratory, 4806-909 Guimarães, Portugal; 3CITAB—Centre for the Research and Technology of Agro-Environmental and Biological Sciences, University of Trás-os-Montes e Alto Douro, 5000-801 Vila Real, Portugal; 4Centre of Molecular and Environmental Biology (CBMA), University of Minho, Campus of Gualtar, 4710-057 Braga, Portugal; acpdias@bio.uminho.pt; 5Identificação e Pesquisa em Princípios Ativos Naturais—NIPPAN, Faculdade de Farmácia, Universidade Federal de Juiz de Fora, Rua José Lourenço Kelmer, s/n—Campus Universitário, Bairro São Pedro, Juiz de Fora 36036-900, Brazil; ademar.alves@ufjf.br

**Keywords:** experimental osteoarthritis, nociception, comorbid mood-like disorders, microglia, medicinal plants, functional foods, *Baccharis dracunculifolia*

## Abstract

Osteoarthritis (OA) persistently activates nociceptors, leading to chronic pain, which is often accompanied by the comorbid development of emotional impairments (anxiety and depression), an effect associated with microgliosis. *Baccharis dracunculifolia* DC (Asteraceae), a Brazilian edible plant, is an important source of active compounds with anti-inflammatory abilities. Thus, we evaluated its ability to reverse OA-induced nociceptive and emotional-like impairments in osteoarthritic ovariectomized female rats using the kaolin/carrageenan (K/C) model. Four weeks after OA induction, mechanical hyperalgesia was confirmed, and the treatment started. Control animals (SHAMs) were treated with phosphate-buffered saline (PBS), while arthritic animals (ARTHs) either received PBS or *B. dracunculifolia* 50 mg/kg (Bd50) and 100 mg/kg (Bd100), via gavage, daily for five weeks. At the end of the treatment, anxiety-like behavior was assessed using the Open Field Test (OFT), anhedonia was assessed using the Sucrose Preference Test (SPT), and learned helplessness was assessed using the Forced Swimming Test (FST). After occision, microglia were stained with IBA-1 and quantified in brain sections of target areas (prefrontal cortex, amygdala, and periaqueductal grey matter). Treatment with *B. dracunculifolia* extract reversed OA-induced mechanical hyperalgesia and partly improved depressive-like behavior in OA animals’ concomitant to a decrease in the number of M1 microglia. Our findings suggest that *B. dracunculifolia* extracts can potentially be used in the food industry and for the development of nutraceuticals and functional foods**.**

## 1. Introduction

Pain, the main reason for seeking health care around the world, is the most common complaint of osteoarthritic (OA) patients [1]. In addition, OA patients frequently exhibit clinical signs of comorbid emotional impairments, such as depression and anxiety [1,2], which are often overlooked by primary care doctors [3].

Hawker et al. [4] showed that OA pain causes fatigue and disability, which, in turn, leads to depression. In parallel, mood impairments alter central pain processing, and depressed patients experience more pain [3], thus highlighting the existence of a synergistic relationship of negative effects between pain and negative emotions [2,5]. While depression prevalence in primary care patients is approximately 10% [6], it increases by 4-fold in OA patients (40.7%) [3]. Depression is also more prevalent in women with (female/male ratio = 1.5:1) or without (female/male ratio = 2:1) chronic pain, and women report more severe clinical pain and feeling more depressed than men [7]. Importantly, patients suffering from chronic pain often display depression/anxiety [2], with recent assessments showing co-occurrence rates of 18–30% and 56–60% [3].

M1-polarized microglia are capable of generating pro-inflammatory cytokines, reactive oxygen species, and nitric oxide, which implies their potential role in contributing to the dysfunction of neural networks in the central nervous system (CNS) [8]. In contrast, M2-polarized microglia express cytokines and receptors associated with inhibiting inflammation and facilitating the restoration of homeostasis [8,9]. M1 microglia are present in the brains of patients with major depressive disorder, and preclinical studies have shown that their presence in the ventral tegmental area, nucleus accumbens, amygdala, and rostroventromedial medulla contributes to depression progression and severity [5,9,10].

In recent years, the interest in alternative therapies has increased [11,12]. Also, the changing dietary patterns observed in recent decades (the increased intake of highly processed fast foods) and their potential implications for public health [13] have amplified the trend of more health-conscious approaches to diets among consumers, increasing the consumption of foods recognized for their functional attributes—those that confer benefits on one or more physiological functions, contributing to individual health and well-being [14,15].

*Baccharis* species are documented in the literature for their diverse applications, serving as sources of food, fodder for livestock, tools, and plants used for dyeing purposes [16]. Regarding *Baccharis dracunculifolia* DC (Asteraceae), traditional healing practices involve the use of teas produced from the leaves, employed to address issues related to the liver and stomach [17]. Notably, several studies have revealed a connection between the chemical composition of *B. dracunculifolia* and Brazilian green propolis, extensively utilized for general consumption and phytopharmaceutical purposes [18,19]. Chemical substances found in the plant, such as flavonoids and coumaric acid derivatives, are present in Brazilian green propolis, as demonstrated in studies by Park et al. [20] and Kumazawa et al. [21]. This correlation underscores the direct influence of the botanical source *B. dracunculifolia* on the composition and potential therapeutic properties of Brazilian green propolis [18]. Moreover, infusions and decoctions of *B. dracunculifolia* flowers are commonly used to alleviate inflammatory processes and to manage liver disorders and stomach ulcers [22]. Indeed, this South American plant is an important source of active compounds, showing antimicrobial [23], antigenotoxic, antimutagenic [11], immunomodulatory [24], anti-inflammatory, and free radical scavenging activities [25,26] and could be a source of interesting functional food ingredients.

In this work, we aim to demonstrate that the oral administration of an extract of *B. dracunculifolia* improves OA-induced nociceptive and comorbid emotional impairments while decreasing M1 microglia in brain areas mediating pain and emotions.

## 2. Materials and Methods

### 2.1. Plant Material

*B. dracunculifolia* aerial parts were gathered from the garden of the Faculty of Pharmacy at the Federal University of Juiz de Fora (Juiz de Fora, Brazil). A voucher specimen CESJ 47482 has been deposited at the Leopoldo Krieger Herbarium in the Institute of Biological Sciences at the Universidade Federal de Juiz de Fora. The plant material was air-dried, triturated, and the resulting powder was macerated with a hydroalcoholic solution (8:2 *v*/*v*; Ethanol P.A. Vetec, Sao Paulo, Brazil). The solution was then filtered and concentrated using a rotary evaporator (Buchi^®^ RII, Flawil, Switzerland) under reduced pressure (Buchi^®^ V-700 pump, Flawil, Switzerland) and bath at 50 °C. The extract was subsequently lyophilized (Christ^®^ Alpha 2–4, B. Braun, Tuttligen, Germany) and stored in the dark at room temperature [27].

### 2.2. HPLC-DAD and UPLC-ESI-QTOF-MS Analysis

The *B. dracunculifolia* extract was solubilized in methanol (3 mg/mL; Sigma-Aldrich, Lisbon, Portugal), filtered, and subjected to HPLC-DAD analysis to quantify the major phenolics present, following a previously described method [23]. The analysis was conducted using a Hitachi-Merck ELITE LaChrom system, comprising an L-2130 pump, an L-2200 autosampler, an L-2300 column oven (operating at 30 °C), and an L-2455 diode array detector (Merck-Hitachi, Tokyo, Japan). Chromatographic separation was achieved using a LiChrospher RP-18 end-capped HPLC Column (5 μm particle size, 250 × 4.6 mm, Merck, Darmstadt, Germany), with water/formic acid (99.9:0.1; solvent A; Sigma-Aldrich, Lisbon, Portugal) and methanol/formic acid (99.9:0.1, solvent B; Sigma-Aldrich, Lisbon, Portugal) as mobile phases. The elution involved a gradient, starting with 10% of solvent B and maintaining the same concentration for 3 min. It then increased to 90% B at 35 min, maintaining 90% B until 45 min, decreasing to 10% B at 50 min, and maintaining 10% B at 60 min. Spectral data were collected in the wavelength range of 245–530 nm, and chromatograms were recorded at 260, 280, and 350 nm. Caffeoylquinic acids and flavonoid derivatives were quantified using the external standard method, expressed as equivalents of chlorogenic acid and kaempferol pure standards (Sigma, Barcelona, Spain), respectively, as described by Dias et al. [27].

For a more precise identification and confirmation of the compounds present in *B. dracunculifolia*, UPLC-MS analysis was conducted using an Acquity UPLC system equipped with a binary pump, in-line degasser, and an autosampler coupled to electrospray ionization quadrupole time-of-flight tandem mass spectrometer (ESI-Q-TOF/MS) (Waters Corp., Milford, MA, USA). *B. dracunculifolia* extract was dissolved in methanol (Sigma-Aldrich, Lisbon, Portugal) and filtered using a 0.22 µm filter before analysis. Chromatographic separation was achieved using a Waters Acquity UPLCTM BEH C18 column (100 mm × 2.1 mm, particle size of 1.7 µm), connected with a guard column (Vanguard 5 × 2.1 mm). The mobile phase consisted of water (solvent A) and acetonitrile (solvent B; Sigma-Aldrich, Lisbon, Portugal), both containing 0.1% formic acid (Sigma-Aldrich, Lisbon, Portugal). The ESI-MS spectrometer (Micromass, Manchester, UK) was operated in both negative and positive modes with a scan range from m/z 100 to 1000. The source temperature was 120 °C; the desolvation temperature was 450 °C; the capillary voltage was set at 2.5 kV; and the cone voltage was 40 V. Nitrogen was used as desolvation gas (800 L/h) and cone gas (50 L/h). Collision energy was applied in ramp mode, ranging from 15 to 30 eV. For accurate mass measurements, data were centroided during acquisition, and 200 pg/mL of leucine–enkephalin (*m*/*z* 565.2771) dissolved in acetonitrile/0.1% formic acid (50:50, *v*/*v*) was continuously infused as an external reference (LockSpray™) into the ESI source with automatic mass correction enabled. Data were processed using Chromalynx™ application manager with MassLynx™ 4.1 software (Waters Corp., Milford, MA, USA). Compound identification was based on MS spectra, and data were obtained by QTOF–MS analysis, compared with information available in the literature and several online databases (ChemSpider, MassBank, and Spectral Database for Organic Compounds).

### 2.3. In Vivo Assay of the Pharmacological Potential of B. dracunculifolia

#### 2.3.1. Ethical Considerations and Handling

The experimental protocol received approval from the Institutional and National Ethical Commission (DGAV 23875/2019) and adhered to the European Community Council Directive 2010/63/EU regarding the use of animals for scientific purposes. Every effort was made to minimize animal suffering and to utilize only the necessary number of animals to generate reliable scientific data. The animals, specifically ovariectomized females, albino Wistar rats (*n* = 6 in each group; Charles Rivers, Barcelona, Spain), were housed in a controlled temperature environment at 22 °C, with 55% relative humidity, and a 12-h light/dark cycle (lights off at 8 pm). They were kept in standard polycarbonate cages (45.4 × 25.5 × 20 cm) with a maximum of three animals per cage and provided ad libitum access to water and food (F0021; BioServ, Flemington, CA, USA).

General health parameters were assessed weekly, and all animals were handled daily by the researcher. On the day of the experimental sessions, the animals were allowed to acclimate to the experimental room for an hour to habituate to the surroundings. All protocols were executed during the light phase of the cycle, with the exception of the Sucrose Preference Test.

#### 2.3.2. Anesthesia and Euthanasia

For ovariectomy and induction of experimental OA, animals were anesthetized with ketamine (0.75 mg/kg, Merial, Lyon, France) and medetomidine (0.5 mg/kg, Esteve Veterinaria, Léon, Spain) intraperitoneally (i.p.) [28]. The size of the pupils, general muscle tone, and absence of nociceptive withdrawal reflexes were monitored to assess the depth of anesthesia. Body temperature was maintained within the physiological range using a warming blanket. Anesthesia was reversed using atipamezole hydrochloride (1 mg/kg, Pfizer, Oeiras, Portugal). For euthanasia, animals received a lethal dose of sodium pentobarbital (200 mg/mL, Ceva, Oeiras, Portugal).

#### 2.3.3. Ovariectomy

Ovariectomy was performed to simulate post-menopause in women [29]. Animals were positioned in ventral recumbency, and the lumbar spine area was shaved to remove hair, followed by cleaning with chlorohexidine (B. Braun). A longitudinal dorsal incision (1–2 cm) was made over the lumbar vertebrae, and the skin on each side was separated from the underlying muscle using blunt-end forceps. The ovaries, embedded in the fat pads, were gently withdrawn, exposing the ovary, oviduct, and part of the uterus. A suture (4/0 Mersilk Soie Perma-hand (W501) Ethicon, Tuttligen, Germany) was tightly tied around the cranial portion of the uterus and uterine vessels. The exposed organs were removed, and the caudal portion of the uterus was reintroduced into the abdominal cavity. Muscle and skin were sutured (muscle: 3/0 Mersilk absorbable (W571H); skin: 2/0 Monocryl absorbable (W3448), Ethicon, Tuttligen, Germany), and the sutured area was cleansed with chlorhexidine (B. Braun, Sempach, Switzerland).

#### 2.3.4. OA Induction

OA induction was carried out following the procedures outlined by Pinto-Ribeiro et al. [30] and Amorim et al. [31]. A mixture of 3% carrageenan–kaolin (Sigma-Aldrich, St Louis, MO, USA) dissolved in sterile saline solution (0.9% NaCl, pH 7.2, Unither, Amiens, France) was injected (0.1 mL) into the right knee joint. Control animals (SHAMs) were injected with 0.1 mL of saline solution. Subsequently, the injected knee was flexed and extended ten times.

#### 2.3.5. Drug Preparation and Administration

The *B. dracunculifolia* powdered extract was dissolved in 10 mM PBS (Applichem, Panreac, Barcelona, Spain) at pH 7.4 to achieve a final concentration of 50 mg/mL. The solution was placed on an ultrasound bath (Sonicator Branson 2510, Emerson Corporate, London, UK) for 30 min and subsequently stored in the dark at room temperature.

The concentration of the extract administered to each animal was adjusted weekly based on their body weight, ensuring that the final volumes administered (1 mL for the Bd50 group and 1.5 mL for the Bd100 group) contained 50 and 100 mg of extract per kg of rat, respectively [17]. Administration occurred daily for 5 weeks through gavage. SHAM and OA animals were administered the vehicle solution.

### 2.4. Behavioral Analysis

#### 2.4.1. Pressure Application Measurement (PAM)

This method facilitated the accurate behavioral measurement of primary mechanical hypersensitivity in rodents experiencing chronic inflammatory joint pain. A force range of 0–1500 g was directly applied to the affected joint [32]. With the animal securely restrained, an increasing force was gradually exerted across the joint until it reached the maximum force necessary to elicit a response, such as paw withdrawal, vocalization, or wriggling movements. The results correspond to the force peak applied immediately before the response and are expressed in grams force (gf). The test was conducted twice (i) before and (ii) at the end of pharmacological treatment. Measurements were recorded as the limb withdrawal threshold (LWT), measured twice on each knee joint at 1 min intervals. The mean LWTs were calculated per animal. To prevent bias, the analysis of PAM (Ugo Basile, Gemonio, Italy) data was performed as follows:PAM = (LWT after *B. dracunculifolia* treatment − LWT before *B. dracunculifolia* treatment).

#### 2.4.2. Open Field Test (OFT)

The Open Field Test (OFT) was employed to assess differences in locomotor and anxiety-like behaviors [32]. The OF apparatus comprised a square arena measuring 100 cm (W) × 100 cm (L) × 40 cm (H) placed in a dimly lit room, featuring a light gradient from the center to the periphery. A single rat was positioned in the center of the arena, and its behavior was recorded using a video camera. Anxiety-like behavior was evaluated based on the analysis of the following parameters: (i) number of squares crossed; (ii) number of rearings; and (iii) time spent in the center. Following each test, the area was cleaned with a 10% alcohol solution. Two blind researchers independently scored the recordings.

#### 2.4.3. Forced Swimming Test (FST)

The evaluation of learned helplessness followed a protocol outlined by Rénéric et al. [33]. On the first day, animals underwent a pre-test session lasting 10 min, during which they were individually placed in a glass cylinder measuring 50 cm (H) × 29 cm (W), filled with water (21.5 ± 1.5 cm, 24 ± 0.5 °C), for 10 min. Twenty-four hours later, the animals were again placed in the cylinder for 5 min, and the session was recorded using a video camera. Behavioral measures were defined as follows: (i) immobility—lack of movement of the whole body; (ii) struggling—vigorous movements with the front paws in and out of the water; (iii) swimming—maintaining active swimming motions; (iv) latency to immobility—time taken by the animal to stop moving for the first time [32]. Two blind researchers scored the test sessions. Learned helplessness was characterized by an increase in immobility time at the expense of the time spent swimming/struggling and a decrease in the latency to immobility.

#### 2.4.4. Sucrose Preference Test (SPT)

Anhedonia, assessed as a reduction in Sucrose Preference Test (SPT), was measured using a protocol adapted from Rodrigues-Fonseca et al. [34]. Prior to testing, animals were exposed to the sucrose solution (Labchem, Laborspirit, Portugal) for 2 h. On the testing night, animals were housed individually, and two pre-weighted bottles were provided—one was a 3% sucrose solution, and the other was sterilized water. After 12 h, the bottles were removed and weighed. Sucrose preference was calculated using the following equation:Sucrose_preference = [sucrose_intake/(sucrose + water)_intake] × 100.

Results were adjusted based on the body weight of the animals.

### 2.5. Histological Processing and Analysis of the Internal Organs

At the conclusion of experimental procedures, the internal organs (thymus, lung, heart, spleen, liver, kidneys, and adrenals) were sampled, and the brain was excised and stored in 4% paraformaldehyde (PFA) (DAC, Applichem, Barcelona, Spain) for further processing.

Histopathological samples were stained with hematoxylin and eosin [35]. The samples underwent dehydration, followed by three washes with xylene and immersion in paraffin (Thermo Scientific, Leicestershire, UK). Blocks were sectioned into 4 μm sections and mounted on a micro-slide glass (Superfrost Plus, Thermo Scientific, Leicestershire, UK), submerged in hexane (S.T. Chemical, Tokyo, Japan) and then in an ethylene/propylene mixture (Clear Plus, Falma Co., Tokyo, Japan), followed by three washes in 100% ethanol. The slides were transferred to a xylene solution (C8H10, Carlo Erba, Cedex, France), washed twice in ethanol, transferred to a 96% ethanol solution, and washed with tap water. The slides were then immersed in hematoxylin Harris solution (Millipore Corporation, Burlington, MA, USA), washed in tap water, immersed in a 0.5% ammonia solution (Sigma Aldrich, Darmstadt, Germany), rewashed in tap water, immersed in 96% ethanol, followed by immersion in eosin Y solution (Thermo Scientific, Cheshire, UK). Slides were then dehydrated with 96% absolute ethanol, immersed in xylene, followed by an ethylene/propylene mixture, and finally mounted with Entellan (Merck, Rahway, NJ, USA). After drying, all slides were examined under a light microscope (Olympus BX61, Olympus Co. Ltd., Tokyo, Japan).

### 2.6. Brain Processing and Immunohistochemistry for IBA-1

Following occision, the right brain hemisphere was excised, preserved in 4% PFA (DAC, Applichem, Barcelona, Spain) for a week, placed in a 20% sucrose solution (Sigma-Aldrich, Lisbon, Portugal) for 48 h, and sectioned using a vibratome (Leica, Carnaxide, Portugal). Coronal sections (50 µm thick) were stored in 12-well plates in 0.1M PBS at 4 °C [35]. Brain sections were collected serially in three sets, and one set was used to assess microglial activation through IBA-1 immunohistochemistry. For IBA-1 immunohistochemistry [36], sections were washed thrice with 0.1M PBS (Sigma Aldrich, Sintra, Portugal), incubated in a 3.3% hydrogen peroxide solution (Carlo Erba, Emmendingen, Germany) in 0.1M PBS, and washed once with 0.1M PBS and twice with PBS/T 0.3%. Subsequently, sections were incubated in 2.5% fetal bovine serum (FBS; Biochrom, Cambridge, UK), followed by overnight incubation with the primary antibody IBA-1 (Abcam, Cambridge, UK) at room temperature. The next day, sections were washed in PBS/T (thrice), incubated in biotinylated swine anti-rabbit secondary antibody (Abcam, Cambridge, UK), and washed in PBS/T (thrice), followed by incubation with avidin–biotin complex (Vector Laboratories, Peterborough, NH, USA). Then, sections were washed with PBS/T (thrice), PBS (thrice), and thrice in Tris buffer (Ultrol Gade, Calbiochem, San Diego, CA, USA), and stained with a diaminobenzidine solution (Sigma Aldrich, Lisbon, Portugal) in Tris buffer. After the reaction, the sections were washed with Tris (twice) and PBS (twice). Sections were mounted on microscope glass slides (Superfrost Plus), dehydrated in increasing concentrations of ethanol (20%, 40%, 70%, 90%, and absolute ethanol), immersed in xylene (C8H10), and finally mounted with Entellan (Merck).

M1 microglial cells were quantified in cingulate (CC), prelimbic (PLC), and infralimbic (ILC) cortices, in the amygdala (AMY) and the dorsolateral (DlPAG) and lateral (LPAG) areas of the periaqueductal grey matter (PAG) based on their phenotype [37]. A minimum of 3 sections per animal were randomly sampled and analyzed using the optical fractionator method (100 × 100 μm) with Stereo Investigator 10 software (Microbrigthfield Bioscience, Magdeburg, Germany). The analysis was conducted using a video camera (Sony, 3CCD, Exwavw HAD) coupled to a microscope (Axioplan2 Imaging, Zeiss, Olympus Co., Ltd., Tokyo, Japan).

### 2.7. Experimental Design

Upon arrival, animals were randomly distributed three per cage and quarantined. One week later, they were transferred to the housing facility, and daily handling was conducted for 15 min over a period of 2 weeks. The animals underwent ovariectomy surgery and OA induction and were then subdivided into the ARTH, ARTH—Bd50, and ARTH—Bd100 groups.

Four weeks after OA induction, the treatment with *B. dracunculifolia* extract was initiated, and animals received daily treatment through gavage for 5 weeks. In the last week of treatment, mechanical hyperalgesia (PAM), anxiety- (OFT), and depressive-like behaviors (FST and SPT) were assessed. Subsequently, the animals were euthanized using pentobarbital, and the internal organs and brain were removed for further processing and analysis (Figure 1).

### 2.8. Statistical/DATA Analysis

Statistical analysis was carried out using the GraphPad Prism 5 software (La Jolla, CA, USA). Behavioral data and the quantification of Iba-1 positive cells were analyzed using a one-way ANOVA to compare ARTH, Bd50, and Bd100 groups, followed by Bonferroni’s post hoc test. T-tests were employed to compare SHAM and OA groups. Results are expressed as mean ± standard error (SEM). Correlation analysis was performed between the FST, OFT, and SPT. A significance level of *p* < 0.05 was considered significant.

## 3. Results

### 3.1. B. dracunculifolia Extract Phytochemical Composition

The chromatographic data and the identification of *B. dracunculifolia* compounds were obtained using UPLC-ESI-QTOF-MS in both positive and negative modes. Additionally, HPLC-DAD data were utilized for qualitative and quantitative analysis. Pure standard compounds, such as caffeic acid, coumaric acid, chlorogenic acid, and kaempferol, were employed for quantification when available. In cases where pure standards were not available, caffeoylquinic acids and flavonoids were quantified as equivalents of chlorogenic acid and kaempferol, respectively.

A representative *B. dracunculifolia* HPLC-DAD chromatogram is illustrated in Figure 2. The major compounds present in the *B. dracunculifolia* extract were initially tentatively identified based on typical UV-Vis spectra. Further confirmation of identification was achieved through UPLC-ESI-QTOF-MS, relying on molecular and fragment ion data. Detailed mass ion information and maximum absorption of each peak are provided in Table 1. Compound identification was validated using previously published data (references in Table 1).

### 3.2. Animal Welfare

Statistical analysis revealed no differences in organ weights between SHAM and ARTH groups (Figure 3a: liver, t_(8)_ = 0.5746, *p* = 0.5814; Figure 3c: adrenals, t_(8)_ = 0.3481, *p* = 0.7368; and Figure 3e: kidneys, t_(8)_ = 1.348, *p* = 0.7368). Similarly, no differences were found between OA groups (Figure 3b: liver, F_(3,14)_ = 1.414, *p* = 0.2810; Figure 3d: adrenals, F_(3,14)_ = 0.4035, *p* = 0.6767; and Figure 3f: kidneys, F_(3,14)_ = 0.1666, *p* = 0.8485). The histopathological examination revealed no abnormalities in the internal organs of our experimental animals (Figure 3g–j).

### 3.3. Pressure Application Measurement (PAM)

ARTH animals exhibited a two-fold decrease in LWT compared to SHAMs (Figure 4a: t_(8)_ = 4.164, *p* = 0.0031). A comparison between OA groups showed that LWT varied among experimental groups (Figure 4a: F_(2,14)_ = 4.566, *p* = 0.0335). Post hoc tests revealed that Bd50 treatment significantly decreased LWT (*p* < 0.05).

### 3.4. Open Field Test (FST)

The OFT results are summarized in Figure 4 and Figure 5. Significant differences were observed between SHAM and ARTH animals, as the total number of squares crossed (Figure 4b: t_(8)_ = 6.915, *p* = 0.0001) and entrances in the central area (Figure 4c: t_(8)_ = 5.657, *p* = 0.0005) were decreased in the ARTH group. Furthermore, ARTH animals spent more time immobile (Figure 4d: t_(8)_ = 5.516, *p* = 0.0006). *B. dracunculifolia* treatment did not improve ARTH animals’ performance in the OFT (Figure 5b: number of crossed squares: F_(2,14)_ = 1.194, *p* = 0.3365; Figure 5c: number of entrances in the area center: F_(2,14)_ = 1.311, *p* = 0.3056; and Figure 5d: immobility time: F_(2,14)_ = 1.245, *p* = 0.227).

### 3.5. Forced Swimming Test (FST)

The t-test revealed significant differences between the performance of the SHAM and ARTH groups. In ARTH animals, latency to immobility (Figure 4f: t_(8)_ = 2.641, *p* = 0.0297) and the time spent active (Figure 4g: swimming: t_(8)_ = 8.285, *p* < 0.0028, and Figure 4h: struggling: t_(8)_ = 4.260, *p* = 0.0028) were decreased, while immobility (Figure 4e: t_(8)_ = 3.858, *p* = 0.0048) was increased. The administration of *B. dracunculifolia* did not reverse OA-induced emotional impairments (Figure 5f: latency: F_(2,14)_ = 0.8387, *p* = 0.4561; Figure 5e: immobility time: F_(2,14)_ = 1.054, *p* = 0.3786; Figure 5g: swimming: F_(2,14)_ = 3.054, *p* = 0.0847; and Figure 5h: struggling: F_(2,14)_ = 0.3151, *p* = 0.7356).

### 3.6. Sucrose Preference Test (SPT)

In the SPT, the ARTH group exhibited a decreased preference for the sweet solution compared with the SHAM group (Figure 4i: t_(8)_ = 2.690, *p* = 0.0275). Interestingly, treatment with *B. dracunculifolia* significantly altered the results between experimental groups (Figure 5i: F_(2,14)_ = 44.85, *p* < 0.0001), with post hoc tests showing a reversal of anhedonia.

### 3.7. Correlation Data

In linear regression, a moderate level of agreement between the OFT and FST was observed (Figure 6a: *R* = 0.5699; *p* = 0.033; *n* = 13). The correlations between SPT and FST and between SPT and OFT were non-significant (Figure 6b: *R* = −0.1894; *p* = 0.4680; *n* = 13) or negligible (Figure 6c: *R* = 0.2301; *p* = 0.3820; *n* = 13), respectively.

### 3.8. Immunohistochemistry Staining

ARTH animals displayed an increased number of Iba-1-positive cells compared to the SHAM group in the PFC (Figure 7a: IL: t_(6)_ = 4.386, *p* = 0.0046; Figure 7b: PrL: t_(6)_ = 3.573, *p* = 0.0117; and Figure 7c: Cg: t_(6)_ = 8.199, *p* = 0.0002), AMY (Figure 7g: t_(6)_ = 5.528, *p* = 0.0015), and PAG (Figure 7h: DlPAG: t_(6)_ = 9.788, *p* < 0.0001; Figure i: LPAG: t_(6)_ = 15.10, *p* < 0.0001).

The number of Iba-1-positive cells in the PFC of the OA animals varied with the treatment (Figure 7d: IL: F_(2,11)_ = 11.09, *p* = 0.0037; Figure 7e: PrL: F_(2,11)_ = 16.75, *p* = 0.0009; and Figure 7f: Cg: F_(2,11)_ = 43.65, *p* < 0.0001). Identical results were observed for the AMY (Figure 7j: F_(2,11)_ = 21.30, *p* = 0.0004) and the PAG (DlPAG, Figure 7k: F_(2,11)_ = 9.788, *p* < 0.0001, and LPAG, Figure 7l: F_(2,11)_ = 45,25, *p* < 0.0001). Post hoc tests showed treatment with *B. dracunculifolia* significantly decreased the number of Iba-1 immunoreactive cells in all studied areas.

## 4. Discussion

In this study, we demonstrated that K/C-induced OA induces pain and the development of nociceptive, anxiety- and depressive-like behaviors in rats, primarily attributed to the increased presence of M1 microglia in brain areas associated with pain modulation and emotions. The observed behavioral enhancements following *B. dracunculifolia* treatment correlated with a reduction in the number of supraspinal M1 microglial cells. This underscores the potential of *B. dracunculifolia* extract consumption for controlling pain and associated mood disorders in OA. These findings suggest the extract’s viability as a functional food or an additive to enrich common food products.

### 4.1. B. dracunculifolia Extract Phytochemical Composition

The quantification of phenolics in *B. dracunculifolia* was conducted using HPLC-DAD and ULPC-MS (Table 1). The phytochemical analysis of *B. dracunculifolia* extract revealed its richness in phenolic compounds, commonly associated with antioxidant and anti-inflammatory properties [18,43]. In this study, the phytochemical analysis of the ethanolic extract from *B. dracunculifolia* aerial parts indicated the presence of several caffeoylquinic acids, notably 3,4-dicaffeoylquinic acid and 4,5-dicaffeoylquinic acid, which were present in higher concentrations. Additionally, some methoxyflavonoids (aromadendrin 4-methyl ether and kaempferide) and kaempferol were also identified. The primary compounds identified in our *B. dracunculifolia* extract align with other studies [17,20,21]. Moreover, caffeoylquinic acids predominated, constituting up to 88% of the phenolics present in the extract.

Plant phenolics, particularly caffeoylquinic acids, can synergistically or additively protect against damage induced by free radicals during oxidative stress [44,45]. They act as scavengers of reactive oxygen and nitrogen species, potentially reducing the risk of chronic diseases in humans [44]. These compounds offer various health benefits, including anti-inflammatory [46], anti-carcinogenic [44,47], anti-mutagenic [48,49], neuroprotective [43,50], and glucose-lowering effects [51,52], and anti-obesity properties [52]. Notably, several studies have demonstrated the antinociceptive effect of caffeoylquinic acids [53,54]. Additionally, aromadendrin 4-methyl ether and kaempferide, also present in *B. dracunculifolia* extract, exhibit anti-inflammatory, antioxidant, and gastroprotective bioactivity [42]. Numerous studies have shown the antinociceptive, anti-inflammatory, and antioxidant effects of kaempferol and its derivates [55,56,57]

### 4.2. Animal Well-Being

In a prior study [11], the primary clinical manifestations of *B. dracunculifolia* toxicity included lethargy, reduced locomotor activity, and decreased exploratory behavior. Given that our animals had already exhibited these impairments, we chose to conduct a histopathological analysis of internal organs to evaluate toxicity [58]. As anticipated, considering our concentrations were 20 and 40 times lower than those reported by Rodrigues et al. [11], no abnormalities were observed in our *B. dracunculifolia-treated* animals when compared to controls. Notably, a common complication associated with oral administration via gavage is the perforation of the esophagus or stomach; however, we confirmed the success of intragastric administration.

### 4.3. Treatment with B. dracunculifolia Decreases Mechanical Hyperalgesia

The K/C model induces mechanical hyperalgesia shortly after induction and is often employed for evaluating potential antinociceptive drugs [59]. Pain originating from muscles or joints is diffuse, longer-lasting, and more unpleasant, promoting a patient’s desire to seek healthcare [59]. Currently, joint replacement is the sole treatment for OA [60]. However, individuals undergoing total knee replacement face various complications, including infection, periprosthetic fracture, symptomatic implant loosening, and implant wear leading to mechanical failure [60]. These complications significantly diminish the benefits of total knee replacement and often necessitate surgery revision.

There is no single pharmacological treatment for OA, and patients typically undergo multiple treatments with nonsteroidal anti-inflammatory drugs (NSAIDs), analgesics such as acetaminophen, or narcotic medications [60], all of which exhibit important negative side effects [60,61]. Other treatments include intra-articular knee injections of corticosteroid or hyaluronic acid (HA) [61,62], which Briem et al. [63] have shown to provide short-term symptom relief but may increase OA progression in the long run. Non-pharmacological therapies for knee osteoarthritis were found to decrease pain but not OA progression [64].

Daily treatment with *B. dracunculifolia* extracts partially (100 mg/kg) or completely (50 mg/kg) reversed mechanical hyperalgesia, and, importantly, with no side effects in our animals. Our data align with the results from Santos et al. [65] demonstrating that *B. dracunculifolia* extract (50–400 mg/kg, p.o.) inhibited nociceptive responses in formalin-induced nociception and produced a long-lasting anti-hypernociceptive effect in acute inflammatory pain.

Medicinal plants offer several benefits to human health, in part due to their content in phenolic compounds that display various properties, including antioxidant, anti-inflammatory, and anti-carcinogenic effects [65,66]. Agents derived from plants, such as flavonoids, terpenes, quinones, catechins, alkaloids, and anthocyanins, which can modulate the expression of pro-inflammatory cascades [64,66] show potential against arthritis-induced joint impairments but require extensive research in preclinical and clinical settings to prove their usefulness [64].

### 4.4. B. dracunculifolia Treatment Does Not Alter Anxiety-like Behavior

Mood disorders, including depression and anxiety, are highly prevalent in patients suffering from chronic pain [2,36]. However, they are often overlooked by general practitioners, especially in OA patients [36]. While patients with mood disorders report higher levels of pain and are more prone to developing chronic widespread pain, the mechanisms underlying the comorbidity between chronic pain and mood disorders remain unclear [67].

Our arthritic animals displayed an anxious-like phenotype consistent with decreased exploratory behavior, distance traveled, and the number of entries in the central area, along with increased immobility in the OFT. Although ameliorated, *B. dracunculifolia* administration did not reverse the anxious-like phenotype. This result contrasts with a report showing that rats with chronic mild stress receiving propolis (50 mg/kg) exhibited increases in the percentage of central quadrants crossed, as well as time spent in the center and grooming [68]. It is possible that our extract does not display anxiolytic activity with the concentrations used herein, or that the composition between *B. dracunculifolia* extracts and propolis varied. Notably, *B. dracunculifolia* is one of the main sources of propolis, and its caffeoylquinic acid and apigenin have been shown to improve anxiety-like behavior [69].

OA patients often display motor impairments due to the degradation of joint/periarticular structures [70]. Accordingly, our OA animals also showed some impairments in locomotor ability (OFT), which may bias the results of the behavioral tests that require active movement, such as those assessing anxiety-like behavior. Therefore, the anxiolytic potential of the *B. dracunculifolia* extract should be tested using an anxiety animal model, such as the chronic unpredictable mild stress model, that does not induce locomotor deficits.

### 4.5. B. dracunculifolia Treatment Reversed Depressive-like Behavior

Previous studies suggest that the time frame may be critical for modeling neuropathic pain-induced affective impairments, with the expression of anxiety-like phenotypes requiring four weeks to develop and preceding the observation of depressive-like behavior, which was only noted six weeks post-induction [67]. The extension of our model to five weeks allowed us to demonstrate the development of both anxiety- and depressive-like behaviors. *B. dracunculifolia* treatment increased sucrose preference, especially at the dose of 100 mg/kg, demonstrating a clear antianhedonic effect of our treatment.

Anhedonia is related to the motivational aspects of major depressive disorder, and its evaluation has minimal requirements for motor coordination; thus, our findings are less likely to be confounded by potential motor deficits. Our data align with Reis et al. [68], who reported that propolis extracts exerted antidepressant-like effects in different animal models of chronic stress, and Lee et al. [71], who demonstrated propolis antidepressant-like activity in mice subjected to stress. Besides the possibility of motor impairments masking our results concerning learned helplessness (FST), the duration of our treatment might not have been sufficient to demonstrate the effect of *B. dracunculifolia* treatment, as a major limitation of classical antidepressants is the need for 2–4 weeks of continuous treatment to elicit therapeutic effects [72].

### 4.6. OA Animals Display Increased M1 Microglial Cells

Mounting evidence suggests that M1 microglial cells contribute to the plasticity of synaptic transmission in chronic pain states [73,74]. We observed a significant increase in the number of M1 cells in the brain of ARTH animals compared to SHAM, consistent with studies using the monoiodoacetate (MIA) model [73]. Blocking M1 microglial cells has been shown to alleviate pain hypersensitivity in animal models of chronic pain, indicating that M1 microglia contributes to enhanced pain sensitivity and supporting the emerging perspective that central microglia are a key mechanism in the progression of chronic pain phenotypes [5,74,75].

Both concentrations of *B. dracunculifolia* treatment decreased M1 microglia, a response concomitant with reduced pain sensitivity and the reversal of anhedonic-like behavior. As the effects of *B. dracunculifolia* on nociceptive impairments resemble those of antidepressants, it is probable that *B. dracunculifolia* treatment also alters cytokine networks [76] and inhibits proinflammatory cytokines [66,76].

Importantly, M1 microglial cells and neuroinflammation play a significant role in the pathogenesis of psychiatric/neurodegenerative diseases [9,76,77,78]. Thus, by decreasing M1 microglia, *B. dracunculifolia* extract may also prevent the sustained activation of microglial-dependent cascades known to alter neurotransmission, neuronal plasticity, and neurogenesis [9,76,78].

## 5. Conclusions

Our results demonstrate that the oral administration of *B. dracunculifolia* extract can reduce inflammation and modulate microglia M1 in the brain, representing a remarkable potential for application in the food industry and the development of nutraceuticals and functional foods. These findings suggest that incorporating this extract into food products could not only enhance its anti-inflammatory properties but also promote well-being through its antinociceptive and antianhedonic effects. These results present a valuable opportunity for creating innovative functional foods, offering benefits for both consumer health and the food industry, which is continually seeking natural and effective ingredients.

## Figures and Tables

**Figure 1 foods-13-00535-f001:**
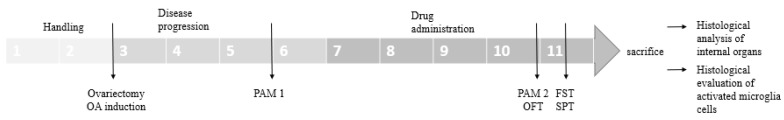
Schematic representation of the experimental time points. Animals were divided into 4 groups (*n* = 6, each): one control group (SHAM) and three experimental arthritis groups—one without treatment (ARTH); one treated with 50 mg/kg extract (ARTH—Bd50); and one treated with 100 mg/kg extract (ARTH—Bd100). Ovariectomy and OA induction were performed 2 weeks after the start of the experimental period. Mechanical hyperalgesia was assessed using the Pressure Application Measurement (PAM) one week before treatment administration and again 4 weeks later. Anxious-like behavior was evaluated using the Open Field Test (OFT) in the 4th week of treatment, and depressive-like behavior was evaluated using the Forced Swimming Test (FST) and Sucrose Preference Test (SPT) in the 5th treatment week. At the conclusion of the experimental period, animals were euthanized, and the internal organs and the brain were excised for histopathological analysis and quantification of activated microglial cells.

**Figure 2 foods-13-00535-f002:**
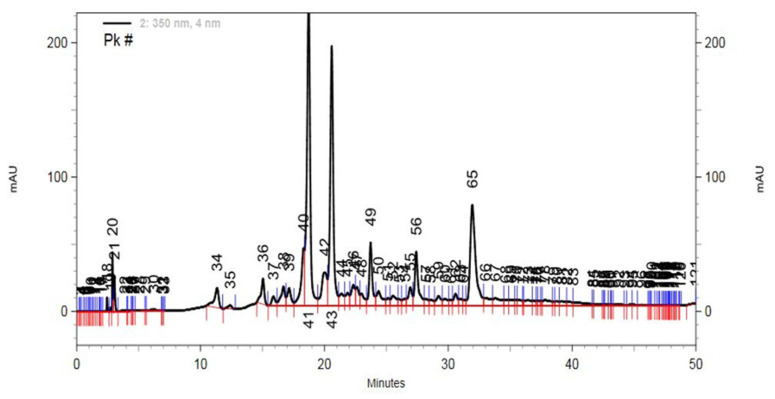
Typical HPLC-DAD chromatogram of *B. dracunculifolia* extract at 350 nm.

**Figure 3 foods-13-00535-f003:**
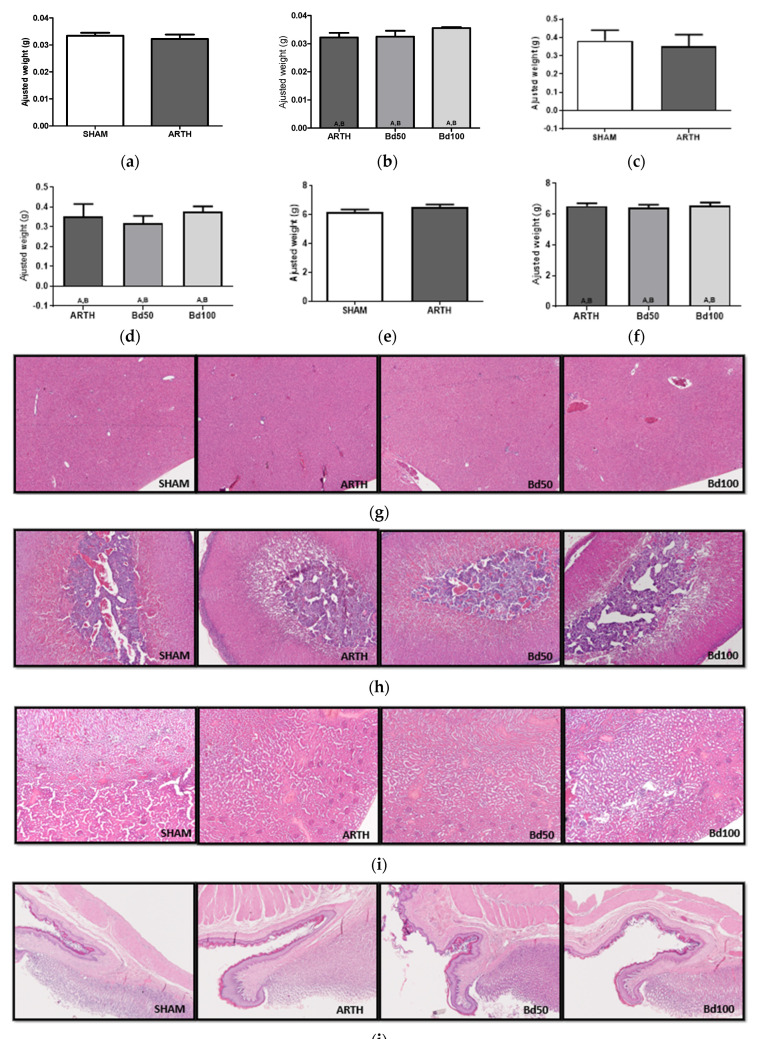
Weight (**a**–**f**) and photomicrographs (**g**–**j**) of internal organs. The weight of internal organs was adjusted to the animal’s body weight. Liver ((**a**)—controls and ARTH; (**b**)—ARTH vs. ARTH—Bd50 and ARTH—Bd100); adrenals ((**c**)—controls and ARTH; (**d**)—ARTH vs. ARTH—Bd50 and ARTH—Bd100); kidneys ((**e**)—controls and ARTH; (**f**)—ARTH vs. ARTH—Bd50 and ARTH—Bd100). No significant differences were found between experimental groups in any of the organs evaluated. Photomicrographs of (**g**) liver, (**h**) adrenals, (**i**) kidneys, and (**j**) inferior esophageal sphincter (photomicrographs: 400× magnification). (SHAM: animals injected with saline in the right knee and treated with PBS; ARTH: animals injected with kaolin/carrageenan in the right knee and treated with PBS; Bd50: animals injected with kaolin/carrageenan in the right knee and treated with 50 mg/kg of Bd extract; Bd100: animals injected with kaolin/carrageenan in the right knee and treated 100 mg/kg of Bd extract). Results are expressed as mean ± SEM; equal means are represented with letters (A, B).

**Figure 4 foods-13-00535-f004:**
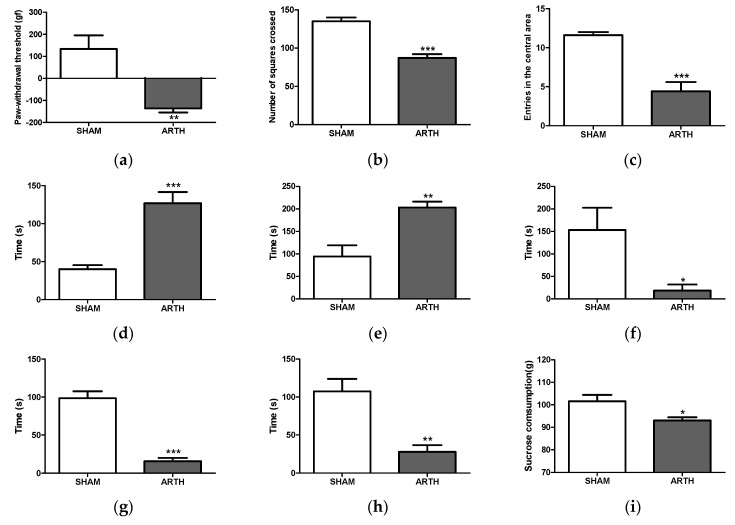
Evaluation of nociceptive and anxiety— and depressive—like behavior in control and arthritic rats. Mechanical hyperalgesia was assessed using the PAM test (**a**), and paw withdrawal latency was significantly decreased in the arthritis (ARTH) group compared to control (SHAM) animals. In the OFT, ARTH animals displayed an anxiety—like phenotype characterized by a decreased exploratory drive with a lower number of squares crossed (**b**) and entries in the central area (**c**), and increased immobility (**d**) compared to SHAM. In the FST, ARTH animals exhibited learned helplessness as they spent more time immobile (**e**) and had increased latency to immobility (**f**). Additionally, ARTH animals spent less time swimming (**g**) and struggling (**h**) than SHAM. ARTH animals displayed an anhedonic—like behavior (**i**), as shown by the significant decrease in the preference for sucrose solution in the SPT. (SHAM: animals injected with saline in the right knee; ARTH: animals injected with kaolin/carrageenan in the right knee; PAM: Pressure Application Measurement; OFT: Open Field Test; FST: Forced Swimming Test; SPT: Sucrose Preference Test). Results are expressed as mean ± SEM. * *p* < 0.05, ** *p* < 0.005 and *** *p* < 0.001.

**Figure 5 foods-13-00535-f005:**
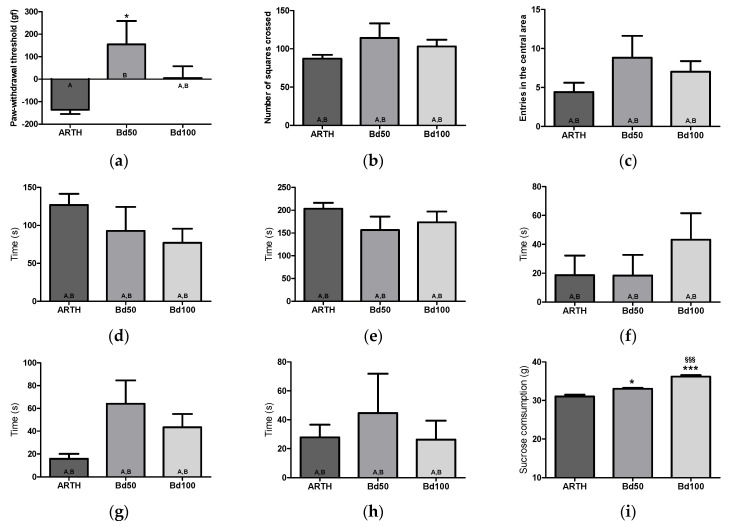
Evaluation of nociceptive and anxiety— and depressive—like behavior in control and Bd-treated rats. In the PAM test, (**a**) mechanical hyperalgesia was partly (Bd100) or completely (Bd50) reversed by the Bd treatment. In the OFT, Bd treatments did not affect anxiety—like behavior (the number of squares crossed (**b**), the number of entries in the central area (**c**), and time spent inactive (**d**). In the FST, learned helplessness was not reversed after Bd treatment (time immobile (**e**); latency to immobility (**f**); time swimming (**g**); and time struggling (**h**). By contrast, Bd treatment reversed anhedonic—like behavior (**i**). (ARTH: animals injected with kaolin/carrageenan in the right knee and treated with PBS; Bd50: animals injected with kaolin/carrageenan in the right knee and treated with 50 mg/kg of Bd extract; Bd100: animals injected with kaolin/carrageenan in the right knee and treated 100 mg/kg of Bd extract; PAM: Pressure Application Measurement; OFT: Open Field Test; FST: Forced Swimming Test; SPT: Sucrose Preference Test). Results are expressed as mean ± SEM. * *p* < 0.05, and *** *p* < 0.001, in comparison with the ARTH group, and ^§§§^ *p* < 0.001 in comparison with the Bd50 group, with equal means represented with letters (A, B).

**Figure 6 foods-13-00535-f006:**
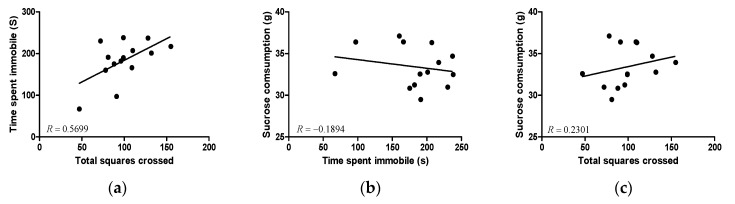
Correlation between the time spent immobile in the FST, the total number of squares crossed in the OFT, and sucrose consumption in the SPT test. (**a**) Correlation between the time spent immobile in the FST and the total number of squares crossed in the OFT; (**b**) correlation between sucrose consumption in the SPT and the time spent immobile in the FST; and (**c**) correlation between sucrose consumption in the SPT and the total number of squares crossed in the OFT. Correlation analysis was conducted using the ARTH—PBS, ARTH—Bd50, and ARTH—Bd100 groups. (ARTH: animals injected with kaolin/carrageenan in the right knee and treated with PBS; FST: Forced Swimming Test; OFT: Open Field Test; SPT: Sucrose Preference Test).

**Figure 7 foods-13-00535-f007:**
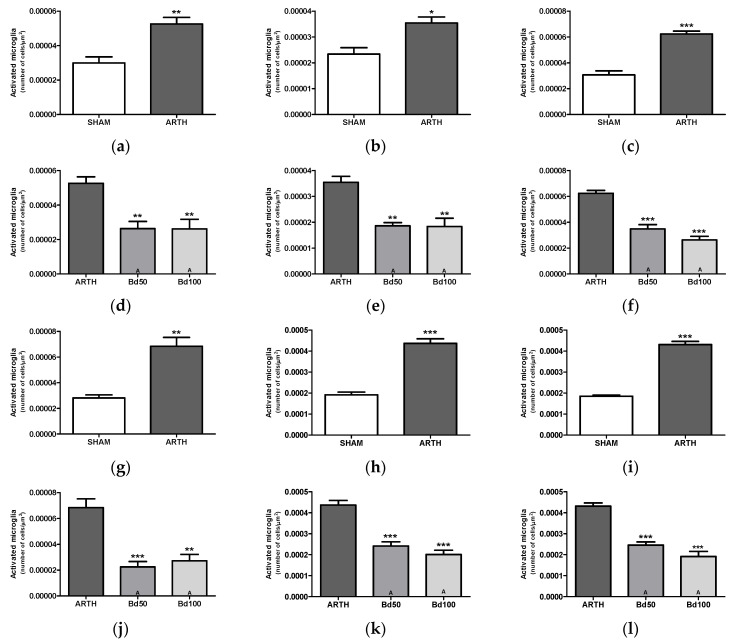
Number of activated microglia cells in the PFC (prefrontal cortex), AMY (amygdala), and PAG (periaqueductal grey matter) was assessed by immunohistochemistry staining with IBA-I. In the PFC, the number of active microglia cells in the IL ((**a**,**d**) infralimbic cortex), PrL ((**b**,**e**) prelimbic cortex), and Cg ((**c**,**f**) cingulate cortex) was increased in ARTH animals, and treatment with Bd reversed microgliosis in the mPFC (**d**–**f**). Similarly, in the AMY, ARTH animals displayed an increased number of active microglial cells (**g**,**j**), and treatment with Bd reversed this effect (**j**). Again, in the PAG, the number of active microglial cells was significantly increased in the DLPAG ((**h**,**k**): dorsolateral periaqueductal grey matter) and LPAG ((**i**,**l**) lateral periaqueductal grey matter), an effect reversed by treatment with Bd ((**k**,**l**)). (ARTH: animals injected with kaolin/carrageenan in the right knee and treated with PBS; Bd50: animals injected with kaolin/carrageenan in the right knee and treated with 50 mg/kg of Bd extract; Bd100: animals injected with kaolin/carrageenan in the right knee and treated 100 mg/kg of Bd extract; PAM: Pressure Application Measurement; OFT: Open Field Test; FST: Forced Swimming Test; SPT: Sucrose Preference Test). Results are expressed as mean ± SEM. * *p* < 0.05, ** *p* < 0.005, and *** *p* < 0.001; equal means are represented with letter (A).

**Table 1 foods-13-00535-t001:** HPLC data and quantification of the major identified compounds in *B. dracunculifolia* extract.

Peak	Compound	*m*/*z* Experimental	ʎ_max_ (nm)	Formula	mg/mg	Reference
34	5-caffeoylquinic acid	353.0877	324	C_16_H_8_O_9_	8.2	[38]
36	Caffeic acid	179.0334	325	C_9_H_8_O_4_	7.5	[39,40,41]
37	Coumaric acid	163.0397	308	C_9_H_8_O_3_	1.8	[39,40,41]
41	3,4-dicaffeoylquinic acid	515.1212	327	C_25_H_24_O_12_	97.6	[39]
42	3,5-dicaffeoylquinic acid	515.1212	327	C_25_H_24_O_12_	17.0	[39,40]
43	4,5-dicaffeoylquinic acid	515.1212	327	C_25_H_24_O_12_	77.6	[40]
49	3,4,5-tricaffeoylquinic acid	677.1562	327	C_34_H_30_O_15_	18.4	[40,41]
55	Aromadendrin 4-methyl ether	301.0708	260, 341	C_16_H_14_O_6_	1.5	[42]
56	Kaempferol	285.0751	264, 365	C_15_H_10_O_6_	7.5	[39,40,41]
65	Kaempferide	300.0634	264, 360	C_16_H_12_O_6_	22.3	[39,40,41,42]

Note: the peak numbers correspond to the ones indicated in Figure 2.

## Data Availability

The data presented in this study are available upon request to the corresponding author. The data are not publicly available due to privacy restrictions.

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
