# Peer review of "Baccharis dracunculifolia DC Consumption Improves Nociceptive and Depressive-like Behavior in Rats with Experimental Osteoarthritis"

_foods, 2024, doi:10.3390/foods13040535_

Round 1
Reviewer 1 Report
Comments and Suggestions for Authors
The manuscript by Laranjeira et al. entitled "Baccharis dracunculifolia..." reports on the ability of a Brazilian edible plant on osteoarthritis-induced nociceptive and emotional-like impairments in ovariectomized female rats using the K/C model. Overall, the manuscript is organized, written well and the study well-designed. There are no major concerns. There are a few minor comments and some minor grammatical points briefly pointed out below.
line 71: has versus as
line 72: increased versus risen
line 93: plant versus pant
Might single sentence be merged in sections such as 2.4.1?
For the diagram of the experimental design, could the figure be moved over to align under the text? This also applies to other graphics.
line 404: kaempferide versus Kaempferide for consistency
In figure 3 c-f, has consideration been given to simplifying the y axis label to mg from g to streamline?
As a note, it is assumed that the figures will be presented on a single page, viz., figure 3, with its legend and all portions visible in the same view. This may simply be an issue of galley versus final typesetting.
Could one-sentence paragraph on lines 544-545 be merged with succeeding paragraph?
line 674: has versus have
line 702: to versus from
Comments on the Quality of English LanguageThe manuscript is very well-written. Use of English language is good.
Author Response
We would like to thank for the reviewer’s effort and dedication in evaluating our manuscript. Please find the detailed responses below and the corresponding revisions/corrections highlighted in the re-submitted files. The points addressed by the reviewer allowed us to considerably improve our work.
Comment #1: Line 71: has versus as; Line 72: increased versus risen.
Response: We thank the reviewer for pointing out these mistakes, as requested, the text was altered and now reads:
Lines 71-73: "In recent years the interest in alternative therapies has increased [11,12]. Also, in recent decades the changing dietary patterns observed (increased intake of highly processed fast foods) and their potential implications for public health (…)”
Comment #2: Line 93: plant versus pant
Response: As suggested by the reviewer the text has now been modified accordingly and now reads:
Line 93: “So, this plant could be a source of interesting food functional ingredients.”
Comment #3: Might single sentence be merged in sections such as 2.4.1?
Response: As suggested by the reviewer, we have moved section 2.5 into section 2.6 (new 2.5) and former section 2.6 into former 2.7 (new 2.6) and adjusted the subsection numbers and titles accordingly. The text now reads:
Lines 261-269: “2.5. Histological processing and analysis of the internal organs
At the end of experimental procedures, the internal organs (thymus, lung, heart, spleen, liver, kidneys and adrenals) were sampled, and the brain excised and stored in 4% paraformaldehyde (PFA) (DAC, Applichem, Barcelona, Spain) until further processing.
Histopathological samples were stained with haematoxylin and eosin [35]. Samples were dehydrated followed by washing thrice with xylene and immersion in paraffin (Thermo Scientific, Leicestershire, UK).”
Lines 284-292: “2.6. Brain processing and Immunohistochemistry for IBA-1
After sacrifice, the right brain hemisphere was excised, preserved in 4% PFA for a week, placed in 20% sucrose solution for 48 h and sectioned in a vibratome (Leica, Carnaxide, Portugal). Coronal sections (50 µm thick) were stored in 12-well plates in 0.1M PBS at 4°C [35]. Brain sections were serially collected in 3 sets, one set was used to evaluate microglial activation through IBA-1 immunohistochemistry.
For IBA-1 immunohistochemistry [36], sections were washed thrice with PBS 0.1M, (Sigma Aldrich, Sintra, Portugal), incubated in 3.3% hydrogen peroxide solution (Carlo Erba) in PBS 0.1M and washed once with PBS 0.1M and twice with PBS/T 0.3%.”
Comment #4: For the diagram of the experimental design, could the figure be moved over to align under the text? This also applies to other graphics.
Response: The alignment of the figures is blocked by features of the journal Foods template, so we are not able to comply with the reviewer’s request. In our experience in the final version of the paper the figures will be align with the text.
Comment #5: Line 404: kaempferide versus Kaempferide for consistency
Response: We appreciate the reviewer for this remark. The text has been revised in accordance with the feedback received.
Comment #6: In figure 3 c-f, has consideration been given to simplifying the y axis label to mg from g to streamline?
Response: As requested by the reviewer, we have updated figures 3c to 3f, that now show the y axis in mg.
Comment #7: As a note, it is assumed that the figures will be presented on a single page, viz., figure 3, with its legend and all portions visible in the same view. This may simply be an issue of galley versus final typesetting.
Response: Indeed, the organization of figure 3, at this moment is set by the template of the journal. We believe that the figure in the final version will be in a single page.
Comment #8: Could one-sentence paragraph on lines 544-545 be merged with succeeding paragraph?
Response: As requested by the reviewer we have adjoined the first two paragraphs of section 4.1, that now reads:
Lines 541-546: “The quantification of phenolics present in B. dracunculifolia was performed by HPLC-DAD and ULPC-MS (Table 1). The phytochemical analysis of B. dracunculifolia extract revealed its abundance in phenolic compounds, commonly associated with antioxidant and anti-inflammatory properties [18,43]. In this study, phytochemical analysis of the ethanolic extract from B. dracunculifolia aerial parts showed the presence of several caffeoylquinic acids (…)”
Comment #9: Line 674: has versus have.
Response: Following the reviewer's suggestions, we have made the requested modifications to the text and now reads:
Line 675 - 676: “Our results show the oral administration of B. dracunculifolia extract has the ability to reduce inflammation and modulate microglia M1 in the brain.”
Comment #10: Line 702: to versus from
Response: In response to the reviewer's feedback, we have incorporated additional revisions into the text and now reads:
Line 703 - 704: “Data Availability Statement: The data presented in this study are available on request to the corresponding author.”
Reviewer 2 Report
Comments and Suggestions for Authors
The Authors did a very good job showing the possibility to use a natural extract as nutraceutical to overcome some collateral symptoms such as pain and depression in osteoarthritis patiets. Following the remark of few errors:
In paragraph 2.3.1. please specify the animals: rats or mice? how many animals for each group?
Paragraph 3.2: what does it means the "t(8)" or "F(3,14)"? and in line 414: I think it is Figure 3 not 2; in the figure 3 you should use pictures of the same area of each organs to make the right comparison. In here are different area and are not comparable;
In addition a general review of th english style and grammar is needed.
Comments on the Quality of English Languagegeneral review of th english style and grammar is needed
Author Response
We would like to thank the reviewer’s effort and dedication in evaluating our manuscript. Please find the detailed responses below and the corresponding revisions/corrections highlighted in the re-submitted files. The points addressed by the reviewer allowed us to considerably improve our work.
Comment #1: In paragraph 2.3.1. please specify the animals: rats or mice? how many animals for each group?
Response: We would like to thank the reviewer for the suggestion. Following their advice, we have incorporated information regarding the species and gender of the animals, along with the respective sample sizes (n) for each group. The text now reads:
Line 159: “The animals (ovariectomized females, albino Wistar rats, n = 6, each group) were kept in a controlled temperature environment, (…)”
Comment #2: Paragraph 3.2: what does it means the "t(8)" or "F(3,14)"?
Response: The t(8) notation refers to the degrees of freedom associated with the t-test distribution. In statistical terms, degrees of freedom represent the number of values in the final calculation of a statistic that are free to vary. In the context of a t-test, it typically involves the sample size minus 1 (n-1), where 'n' is the number of observations. So, t(8) implies a t-distribution with 8 degrees of freedom, and follows the guidelines for reporting statistical results (https://www.statology.org/how-to-report-t-test-results/).
Firstly, in relation to the F(3,14), in section 3.2, it was a typing mistake and should read F(2,14), identically to the remaining results in this section.
In relation to the notation F(2,14) it is used to report results of ANOVA (Analysis of Variance) statistical analysis. The notation F(2, 14) represents: the first number (2) represents the degrees of freedom associated with the number of experimental groups and the second number (14) represents the degrees of freedom associated with the number of animals (https://www.statology.org/how-to-report-anova-results/).
In both the t(8) and F(2, 14) notations, the degrees of freedom are crucial for determining the critical values from the respective distributions. These critical values are used to understand the statistical significance of test results. As a rule of thumb, the larger the degrees of freedom, the closer the t-distribution or F-distribution approximates a normal distribution, allowing for more reliable statistical inferences.
Comment #3: In line 414: I think it is Figure 3 not 2.
Response: We appreciate the correction and have changed the number in the text. Thank you for your input.
Comment #4: in the figure 3 you should use pictures of the same area of each organ to make the right comparison. In here are different area and are not comparable.
Response: As requested by the reviewer, we revisited the slides and took new pictures. We expect the new images showcase the same area in each organ. We also believe we were able the quality of the microphotographs and maintain consistent color- tones.
Comment #5: General review of the english style and grammar is needed.
Response: Thank you for your suggestion. We asked a third party to review the manuscript, considering your evaluation. We believe it increased the consistency of the manuscript and the alterations have been highlighted in blue.